

**Marine carbon dynamics in a coral reef ecosystem of Southern**
**Taiwan**
Pei-Jie Meng[1-3], Chia-Ming Chang[1], Wen-Chen Chou[4-5], Hung-Yen Hsieh[1], Anderson
B. Mayfield[6-7], Chung-Chi Chen[8,1,2]*
[1]Graduate Institute of Marine Biology, National Dong Hwa University, Checheng,
Pingtung 944, Taiwan
[2]Taiwan Ocean Research Institute, National Applied Research Laboratories,
Kaohsiung 852, Taiwan
[3]Department of Oceanography, National Sun Yat-Sen University
Kaohsiung 804, Taiwan
[4]Institute of Marine Environment and Ecology
National Taiwan Ocean University, Keelung 20224, Taiwan
[5]Center of Excellence for the Oceans
National Taiwan Ocean University, Keelung 20224, Taiwan
[6]Coral Reef Diagnostics, Miami, FL 33129, USA
[7]Coral Research and Development Accelerator Platform, Thuwal, Makkah 23955,
Saudi Arabia
[8]Department of Life Science, National Taiwan Normal University
88, Sec. 4, Ting-Chou Rd., Taipei 11677, Taiwan
Running title: $p$CO$_2$ in a coral reef ecosystem
* Corresponding authors: Chung-Chi Chen
Email: ccchen@ntnu.edu.tw
Phone: 886-2-7749-6328; Fax #: 886-2-2931-2904





**ABSTRACT**
The ocean is the planet's largest carbon reservoir and plays a crucial role in regulating
atmospheric $CO_2$ levels, especially in the face of climate change. In coral reef
ecosystems, understanding the carbonate system is critical for predicting and
mitigating the impact of ocean acidification on these vulnerable marine ecosystems,
especially as atmospheric $CO_2$ concentrations continue to rise. This study measured
$pCO_2$ over space and time in Nanwan Bay, a coral reef ecosystem in southern Taiwan,
to identify factors that influence its variation. The results showed that mean surface
water $pCO_2$ values varied seasonally, with values of 393.7 (±10.8), 406.3 (±16.1),
399.2 (±18.6), and 366.9 (±14.5) µatm in spring, summer, fall, and winter,
respectively. These seasonal mean differences ($\Delta pCO_2$) relative to atmospheric $pCO_2$
were 7.7 (±10.8), 29.3 (±16.1), 21.2 (±18.6), and -16.1 (±14.5) µatm, respectively.
These findings suggest that the Nanwan Bay is a highly dynamic coral reef
ecosystem, exhibiting both spatial and seasonal variability in carbon exchange. The
carbonate system parameters of the surface water in this high-biodiversity, sub-
tropical marine ecosystem was influenced not only by seasonal temperature variation
but also by vertical mixing, intermittent upwelling, and biological effects.

**Keywords**: carbon sink, carbon source, coral reef, $pCO_2$, total alkalinity, upwelling



## 1. Introduction


Understanding whether the ocean acts as a carbon dioxide ($CO_2$) sink or source is
crucial in the context of climate change, as it directly affects climate regulation,
ecosystem health, and the effectiveness of mitigation efforts. Oceans absorb about
30% of human-produced $CO_2$ (Ipcc, 2021). If the ocean's ability to absorb $CO_2$
decreases, more $CO_2$ will remain in the atmosphere, exacerbating global warming.
$CO_2$ concentration in marine systems varies between region and over time (Fay et al.,
2021; Sitch et al., 2015; Schimel et al., 2001). For example, high-latitude temperate
regions and coastal seas act as sinks for atmospheric $CO_2$, while subtropical and
tropical coastal seas, estuaries, and coral reefs are generally sources (Borges et al.,
2005; Cai et al., 2003; Frankignoulle et al., 1998; Frankignoulle et al., 1996; Gattuso
et al., 1997; Gattuso et al., 1993; Ito et al., 2005; Ohde and Van Woesik, 1999; Wang
and Cai, 2004; Yan et al., 2011; Bates et al., 2001). The hydrological characteristics of
coastal waters, such as temperature, salinity, upwelling, and mixing, also exhibit
substantial variation, leading to differences in surface water $p CO_2$ even within the
same continental shelf. Furthermore, upwelling areas along the coasts of California
and Oman act as $CO_2$ sinks, whereas those along the coasts of Galicia and Oregon
serve as $CO_2$ sources (Borges and Frankignoulle, 2002; Friederich et al., 2002; Goyet
et al., 1998; Hales et al., 2005). Borges (2005) notes that when estuaries are included



in the gas exchange process, coastal seas worldwide are sources of $CO_2$, but they
become sinks when estuaries are excluded.

Various factors, such as temperature, tides, currents, river discharge, upwelling,

vertical mixing, and biological metabolism, influence $CO_2$ levels in coastal areas (e.g.,
Dai et al., 2009; Chen et al., 2024; Ibanhez et al., 2015). For instance, temperature
changes can directly affect the solubility of $CO_2$ in seawater, with higher temperatures
generally reduce $CO_2$ solubility (Dai et al., 2009). Tides and currents can enhance
vertical mixing, bringing $CO_2$-rich deep waters to the surface and increasing $pCO_2$
levels (Ibanhez et al., 2015; Dai et al., 2009). River discharge introduces fresh water
and organic matter, which can stimulate biological metabolism and further influence
$CO_2$ levels through respiration and decomposition processes (Chen et al., 2024;
Ibanhez et al., 2015). These factors interact and help explain why seasonal variation in
$CO_2$ levels can differ greatly across regions. For example, measurements taken at the
Bermuda Atlantic Time-series station in the Northwest Atlantic from 1996 to 1998
showed that $CO_2$ levels were lowest in winter and highest in summer (Takahashi et
al., 2002). Conversely, data collected from the Kyodo Western North Pacific Ocean
Time-Series station between 1998 and 2000 indicated that $CO_2$ levels were lower in
summer compared to winter (Takahashi et al., 2002).

Coral reef ecosystems are known for their high productivity, biomass, and



efficient carbonate deposition rates. However, due to their structural complexity and
high biodiversity, the carbon dynamics of coral reefs differ substantially from those of
the open ocean. Whether coral reef ecosystems function as net carbon sources (Ware
et al., 1992; Gattuso et al., 1993; Gattuso et al., 1999; Fagan and Mackenzie, 2007;
Lønborg et al., 2019; Yan et al., 2018; Watanabe and Nakamura, 2019; Frankignoulle
et al., 1998) or sinks (Kayanne et al., 1995; Mayer et al., 2018; Suzuki, 1998; Suzuki
and Kawahata, 2004) remain uncertain due to factors like the high spatial and
temporal variability within these ecosystems. Sedimentation rates and anthropogenic
influences further complicate the coral reef carbon balance. Moreover, ocean
acidification could reduce coral calcification rates, diminish their ability to produce
$CO_2$, thereby buffering and reducing the $pCO_2$ levels in seawater (Ries, 2011; Fabry et
al., 2008; Albright et al., 2016). Consequently, predicting changes in seawater carbon
levels in response to oceanographic anomalies remains a challenge.

Nanwan Bay, located at the southernmost tip of Taiwan (Fig. 1), is a semi-

enclosed area between Cape Moubitou and Cape Oluanpi, characterized by high-
biodiversity and abundant fringing coral reefs (Yang and Dai, 1980). Previous studies
of Nanwan Bay have noted that periodic upwelling causes the mixing of upper and
lower seawater layers, facilitating the transfer of nutrients from the deeper waters to
shallow areas (Chen et al., 2005). In most coastal upwelling regions, the ocean



absorbs $CO_2$ from the atmosphere (Hales et al., 2005), a process closely tied to the
increased primary production of phytoplankton in the nutrient-rich waters following
upwelling. Primary productivity in marine ecosystems plays a crucial role in carbon
cycling by driving the fixation of $CO_2$ (Dugdale and Wilkerson, 1989; Murray et al.,
1995). During periods of heightened primary productivity, the increased demand for
carbon can lead to greater uptake of $CO_2$ from seawater, potentially reducing its
concentration (Chen et al., 2004). Interestingly, this phenomenon has yet to be
explored within the coral reef ecosystem of Nanwan Bay.

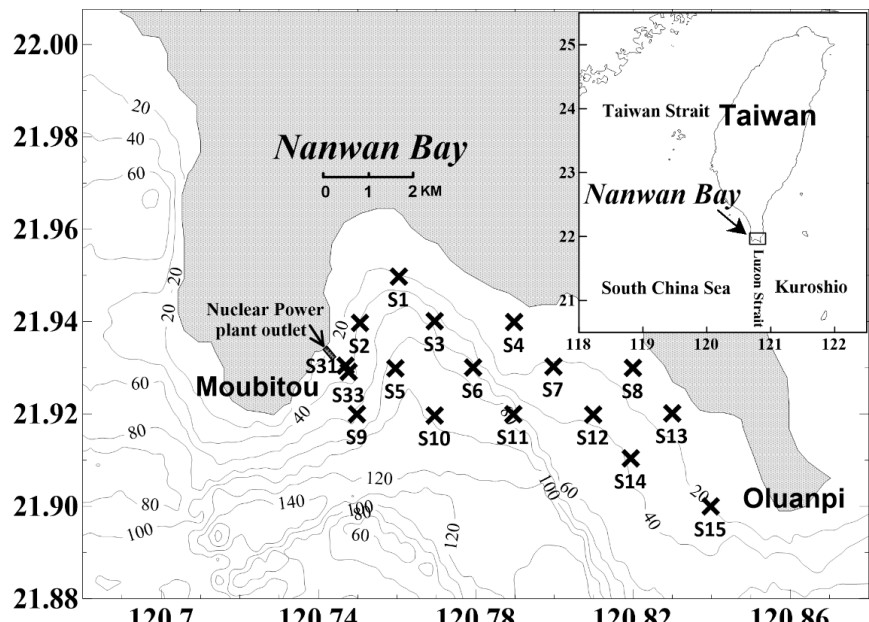


**Fig. 1** Map of sampling stations (marked with "X" with the station number

underneath) in Nanwan Bay, Taiwan, along with contours of depth (m).

The difference between seawater $pCO_2$ ($pCO_2^{seawater}$) and atmospheric $pCO_2$





($p\text{CO}_2^{\text{air}}$) not only serves as a metric but also determines whether a marine system
functions as a source or sink of carbon. In this context, a positive difference, where
$p\text{CO}_2^{\text{seawater}}$ - $p\text{CO}_2^{\text{air}}$ > 0, indicates a carbon source, while a negative difference
signifies a carbon sink. To determine whether Nanwan Bay behaves as a net carbon
source or sink, this study evaluates the role of hydrological conditions and their
potential influence on the carbonate system and $\text{CO}_2$ fluxes. To achieve this, we
conducted a comprehensive analysis of the marine carbonate system across various
spatial and temporal gradients.

**2. Methods**

**2.1 Study site** Nanwan Bay (Fig. 1) is flanked by the Pacific Ocean to the east,

the Taiwan Strait to the west, the Luzon Strait to the south, and the South China Sea
(SCS) to the southwest (Lee, 1999) and covers an area of ~30 km$^2$ (estimated via
Google Earth Pro). Nanwan Bay is among the most diverse marine regions in Taiwan,
which led to its inclusion within Kenting National Park (Meng et al., 2008). The
complex seabed in Nanwan Bay encompasses diverse habitats, including: sandy
beaches, rocky shores, and coral reefs. This area represents the initial point of
interaction between the SCS waters and the warm, highly saline Kuroshio Current.
The water is oligotrophic, and temperatures typically range from 21 to 30°C (with
periodic upwelling events occurring;(Chen et al., 2005). In the course of the





upwelling event, the surface water of Nanwan Bay can drop by >3°C, coupled with a
rise in nitrate concentration exceeding 2 μM, as documented by (Chen et al., 2005).
The bay hosts over 1,200 fish species and more than 200 species of reef-building
coral, making it a significant research focus area for both the reefs and the
anthropogenic stressor regime (Meng et al., 2007). Studies have shown that high
levels of nutrients and suspended solids may have contributed to the decline in coral
cover between 2001 and 2022 (Meng et al., 2008; Chen et al., 2022).
**2.2 Sampling and analysis** The study was conducted across four seasons: spring
(31 March 2011), summer (5 July 2011), autumn (20 October 2011), and winter (22
January 2013), in the area between Nanwan Bay's two capes, Moubitou and Oluanpi.
A total of 17 seawater sampling stations were established, including two (Sts. 31 and
33) located near the outlet of a nuclear power plant (Fig. 1). Temperature and salinity
data were collected using an Idronaut Ocean Seven 304 CTD calibrated against an
International Association for the Physical Sciences of the Ocean seawater standard.
Water samples were collected using Niskin bottles with Teflon-coated inner walls.
Seawater at each station was taken at two to five depths at intervals of 3 to 25 m in
areas shallower than 50 m; extra samples at 65, 80 and/or 100 m were taken for
stations with depths of 65-100 m. Please refer to Table S1 for details on the sampling
times and sampling depths at each station. Notably, all samples were collected



exclusively during daytime. Water samples were immediately analyzed for dissolved
oxygen (DO) content using YSI 52 and YSI 5905 BOD electrodes (factory
accuracy=99.9%). Other water samples were divided into different sample bottles for
additional analyses, including chlorophyll *a* (Chl *a*), pH, and total alkalinity (TA).
One 300-mL amber bottle was pre-inoculated with 0.2 mL of mercuric chloride to
suppress biological activity that could affect TA and other carbonate system
parameters.

For Chl *a* analysis, 1 L of seawater were was immediately filtered through GF/F

filter paper (Whatman, 47 mm) and stored in liquid nitrogen. The Chl *a* retained on
the GF/F filters was determined fluorometrically (Turner Design 10-AU-005; Parsons
et al., 1984).

Seawater pH and total TA were measured using an automated titration system

consisting of a Mettler-Toledo DL53 with a DG-111 electrode. Prior to measurements,
the electrode was calibrated using Merck standard buffer solution (NIST) at 25°C.
The calibration ranges for pH 4, 7, and 10 were set to fall within the range of 176±30
mV, 0±30 mV, and -176±30 mV, respectively (calibration slope of -56 to -59).
Measured pH values were expressed on the NBS scale.

For TA measurements, 40 g of seawater were titrated with 0.1 N HCl at 25°C.

Titration continued until the pH exceeded the end point (~pH 4.4) and then continued



until ~pH 3.0, with the potential change and titration volume recorded. The consumed
volume of HCl was calculated using the Gran (1952) function based on the linear
relationship between titration volume and pH, and TA was obtained by plotting the
consumed volume of HCl. The reference material for experimental quality control
was obtained from Professor Andrew Dickson (Scripps Institute of Oceanography,
USA), and the pH of the reference material was calculated by entering dissolved
inorganic carbon (DIC) and TA data into CO2SYS software ver. 1.02 (Lewis and
Wallace, 1998).

The pH measurement accuracy in this study was ±0.01 units and the TA accuracy

was ±2.7 µmol kg$^{-1}$ (precision=0.12%). $p\text{CO}_2$ and DIC were also calculated with
CO2SYS from measured pH and TA. The dissociation constants of carbonic acid used
were the revised K1 and K2 values from Dickson and Millero (1987) refit from the
values of Mehrbach et al. (1973). Notably, the surface water $p\text{CO}_2$ was estimated
using the average values of samples collected at depths of 1 and 3 m at each station
(Table S1).

**2.3 Calculation of the exchange flux of CO₂ between the ocean and the**

**atmosphere.** The formula for calculating the exchange flux of $CO_2$ between the ocean
and the atmosphere (F$_{GAS}$) was as follows:
$F_{GAS} = k \times K_H \times (p\text{CO}_2^{\text{seawater}} - p\text{CO}_2^{\text{air}})$



where $k$ is the gas exchange rate of $CO_2$ (air-sea gas transfer rate) and $K_H$ is the
solubility of $CO_2$ gas in seawater. The air-sea gas transfer rate, $k$, was obtained from
an empirical formula based on wind speed proposed by Wanninkhof (1992): $k$=0.31 ×
$u^2 \times (Sc/660)^{-0.5}$, where $u$ is wind speed 10 m above sea level (in m/s; data from the
Central Weather Bureau's Oceanic Center-Oluanpi buoy); and $Sc$ (Schmidt number) is
a function of temperature (Wanninkhof, 1992), which can be obtained from the *in situ*
sea surface temperature (T) as follows:
$Sc = 2073.1 - 125.62 \times T + 3.6276 \times T^2 - 0.043219 \times T^3$
The solubility of $CO_2$ gas in seawater ($K_H$), expressed in mol $L^{-1} \cdot atm^{-1}$, was calculated
using the formula developed by Weiss (1974):
$\ln K_H = -58.0931 + 90.5069\left(\dfrac{100}{T}\right) + 22.2940\ln\left(\dfrac{T}{100}\right) + S\left[0.027766 - 0.025888 + \left(\dfrac{T}{100}\right) + 0.0050578\left(\dfrac{T}{100}\right)^2\right]$
Since we did not directly measure $p$CO$_2^{air}$, we used xCO$_2$ data provided by the United
States National Oceanic and Atmospheric Administration (NOAA) from Dongsha
Island
(https://gml.noaa.gov/aftp/data/trace_gases/co2/flask/surface/txt/co2_dsi_surface-
flask_1_ccgg_event.txt). Dongsha Island, located at approximately 20.70°N, is a coral
atoll with a latitude similar to that of Nanwan Bay, and importantly, it shares the
characteristic of being part of a coral reef ecosystem. The dry air xCO$_2$ values were
corrected to 100% humidity, assuming atmospheric pressure of 1 atm, using the





temperature and salinity data recorded at the time of sampling. The resulting $p$CO$_2$$^{air}$
was 386, 377, 378, and 383 μatm on March 31, July 5, and October 18, 2011, and
January 22, 2013, respectively.

The seasonal fluxes across the bay were calculated by multiplying the mean CO$_2$

exchange flux at all stations for each season by the bay's area of ~30 km$^2$.

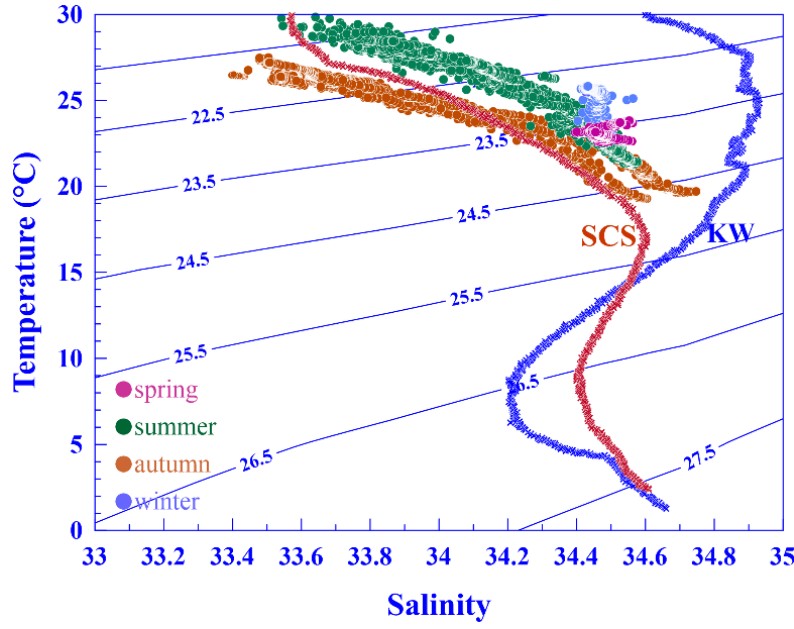


**Fig. 2** Temperature vs. salinity (T-S) diagram at Nanwan Bay, Taiwan in spring,

summer, autumn, and winter. SCS = South China Sea and KW = Kuroshio

Waters.

### 3.  Results and discussion

**3.1 Variation in hydrological parameters.** Both temperature and salinity varied
over time in Nanwan Bay (Fig. 2), with the seasonal variation likely driven by both
the monsoon and SCS circulation patterns as follows. The Kuroshio Current flows





northwards along Taiwan's east coast, with a portion of the Western Philippine Sea
(WPS) water following the Kuroshio and then flowing westward along the northern
SCS shelf (Yuan et al., 2006). Nan et al. (2015) suggested that surface salinity of 34 or
higher is characteristic of the Kuroshio, indicating potential inundation of the
Kuroshio Current into Nanwan Bay during the high-salinity spring period. During
summer, the southwest monsoon dominates, leading to a decrease in the Kuroshio's
influence; the main circulation of the Kuroshio shifts westward to the Luzon Strait,
limiting its intrusion into the northwestern SCS region (Liang et al., 2008). In the
northern SCS region, the southwest-to-northeast circulation pattern prevails during the
monsoon, with most seawater flowing out of the SCS through the Luzon Strait and
converging with the Kuroshio axis, resulting in Nanwan Bay being dominated by the
SCS water mass during the summer. An analysis of temperature and salinity data from
Nanwan Bay, the SCS, and the Kuroshio Current indicates that Nanwan Bay mainly
consists of the SCS water mass during summer and autumn, while during spring and
winter, the water masses are intermediate between the two (Fig. 2). As such, Nanwan
Bay is classified as a mixed water mass area, comprising both SCS and Kuroshio
Current water masses.

During the survey period, there was a clear positive correlation between pH and

temperature. Additionally, pH and TA exhibited significant correlations with salinity





**Table 1.** Correlation matrix of seawater quality variables with correlation coefficients
($r$) for spring, summer, autumn, and winter. Variables include temperature,
salinity, dissolved oxygen (DO), DO saturation (DO%), total alkalinity (TA),
dissolved inorganic carbon (DIC), and pH.

| Spring | Temperature | Salinity | DO | DO (%) | TA | DIC | pH |
|---|---|---|---|---|---|---|---|
| Salinity | 0.35** | | | | | | |
| DO | -0.23 | -0.40** | | | | | |
| DO (%) | 0.02 | -0.32** | 0.89** | | | | |
| TA | 0.04 | 0.03 | 0.19 | 0.11 | | | |
| DIC | -0.17 | -0.04 | 0.25* | 0.11 | 0.92** | | |
| pH | 0.43** | 0.13 | -0.02 | 0.05 | 0.61** | 0.27* | |
| $p$CO$_2$ | -0.28* | -0.07 | 0.03 | -0.02 | -0.45** | -0.09 | -0.96** |


| Summer | Temperature | Salinity | DO | DO (%) | TA | DIC | pH |
|---|---|---|---|---|---|---|---|
| Salinity | -0.96** | | | | | | |
| DO | 0.65** | -0.53** | | | | | |
| DO (%) | 0.90** | -0.81** | 0.90** | | | | |
| TA | -0.82** | 0.81** | -0.52** | -0.72** | | | |
| DIC | -0.91** | 0.84** | -0.68** | -0.87** | 0.91** | | |
| pH | 0.89** | -0.79** | 0.72** | 0.89** | -0.72** | -0.94** | |
| $p$CO$_2$ | -0.22* | 0.07 | -0.45** | -0.38** | 0.23* | 0.51** | -0.64** |


| Autumn | Temperature | Salinity | DO | DO (%) | TA | DIC | pH |
|---|---|---|---|---|---|---|---|
| Salinity | -0.95** | | | | | | |
| DO | 0.88** | -0.85** | | | | | |
| DO (%) | 0.95** | -0.93** | 0.98** | | | | |
| TA | -0.57** | 0.56** | -0.51** | -0.55** | | | |
| DIC | -0.76** | 0.75** | -0.66** | -0.72** | 0.88** | | |
| pH | 0.79** | -0.77** | 0.66** | 0.73** | -0.43** | -0.80** | |
| $p$CO$_2$ | -0.32** | 0.33** | -0.22 | -0.27* | 0.27* | 0.63** | -0.83** |


| Winter | Temperature | Salinity | DO | DO (%) | TA | DIC | pH |
|---|---|---|---|---|---|---|---|
| Salinity | -0.32** | | | | | | |
| DO | 0.43** | -0.26* | | | | | |
| DO (%) | 0.78** | -0.34** | 0.70** | | | | |
| TA | -0.15 | 0.06 | -0.19 | -0.17 | | | |
| DIC | -0.39** | 0.02 | -0.34** | -0.41** | 0.89** | | |
| pH | 0.59** | 0.06 | 0.39** | 0.61** | -0.12 | -0.56** | |
| $p$CO$_2$ | -0.34** | -0.19 | -0.33** | -0.44** | 0.29** | 0.67** | -0.94** |

*: $p \leq 0.05$ and **: $p \leq 0.01$.
during summer and autumn, but such correlations were not evident in spring and



winter (Table 1). It is expected that TA and salinity will covary because the charge
differences between cations and anions in seawater change with salinity. Salinity
generally increases with depth and is influenced by various factors such as rainfall,
evaporation, and freshwater input, which can lead to changes in TA. It is worth noting
that the absence of a major river nearby the study sites, as well as the absence of any

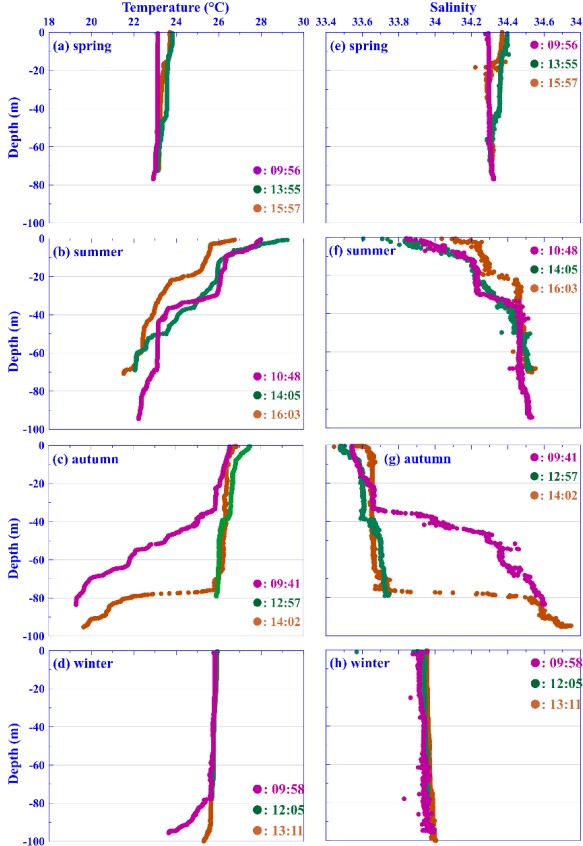


**Fig. 3.** Vertical profiles of temperature and salinity at station S10 in spring (a & e,

respectively), summer (b & f, respectively), autumn (c & g, respectively), and

winter (d & h, respectively) at three sampling times (see Table S1 for details.)

as indicated in each panel.





observed rainfall events one week prior to each survey period, strongly suggests that
freshwater input did not play a significant role in altering TA within this study area.
Moreover, the absence of significant correlations among salinity, TA, and pH in spring
and winter implies that these relationships might be influenced by factors such as
intense vertical mixing or upwelling, which could disrupt the salinity, TA, and pH
vertical profiles. This supposition is further supported by the well-mixed profiles of
salinity and temperature found throughout the water column at station S10 during
spring and winter (Fig. 3a, d, e, & h). Additionally, seawater characterized by low
temperature, low pH, and high salinity observed at station S10 during the spring
suggests that this well-mixed pattern throughout the water column may be primarily
associated with upwelling during this period (Figs. 3 and S1; further details can be
found in the next section).

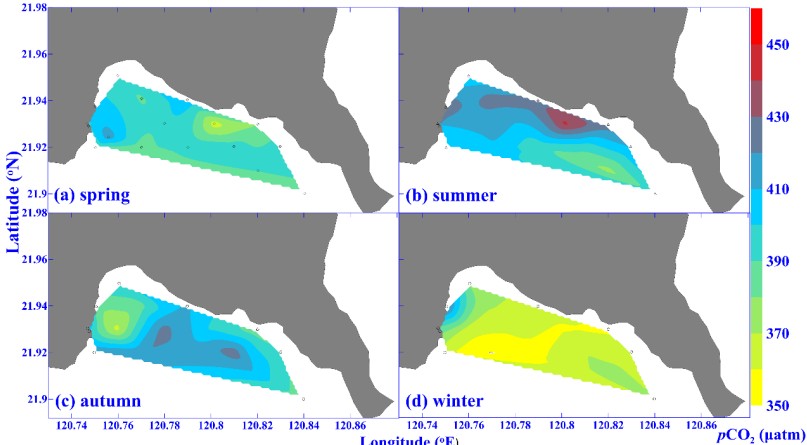


**Fig. 4.** Seasonal variation in sea surface $p\text{CO}_2$ (µatm) in Nanwan Bay, Taiwan in

spring (a), summer (b), autumn (c), and winter (d).



**3.2 Changes in surface water $pCO_2$.** Surface $pCO_2$ levels in Nanwan Bay
ranged from 364–422, 362–448, 350–480, and 345–427 µatm in spring, summer,
autumn, and winter, respectively (Fig. 4). The means values (±SD) across all stations
(N = 17) for each season were 393.2 (±11.6), 411.4 (±19.0), 401.7 (±18.3), and 370
(±17.3) µatm, respectively. The mean surface seawater temperatures during these
seasons were 23.4 (±0.4), 28.8 (±0.8), 27.0 (±1.0), and 26.0 (±0.6)℃, respectively. In
the open ocean, $pCO_2$ levels are primarily influenced by temperature, horizontal
transport and vertical mixing, biological processes, and gas exchange (e.g., Dai et al.,
2009). Due to the mixing of different water masses by monsoons, tides, eddies,
upwelling, and other ocean currents, significant variation in temperature and salinity
of the water column was observed at different times at station S10 (Fig. 3) and in the
carbonate parameter data (Fig. 5d & S1-S4). Similarly, significant diurnal variation in
seawater $pCO_2$ has been reported in another coral reef ecosystem (Yan et al., 2018),
suggesting that more extensive temporal and spatial sampling is needed to accurately
capture the true dynamics of the carbonate system in coral reef environments. During
spring and winter, pronounced mixing was evident, as demonstrated by the straight
vertical profiles in temperature and salinity in Fig. 3a, d, e, and h. Conversely, in
summer and autumn, mixing was less apparent.
According to Lee et al. (1997; 1999a; 1999b), cold-water upwelling occurs with



tidal changes in Nanwan Bay, which increases vertical mixing. The temperature-
salinity-pH-DO diagram of station S1 illustrates that throughout the entire upwelling

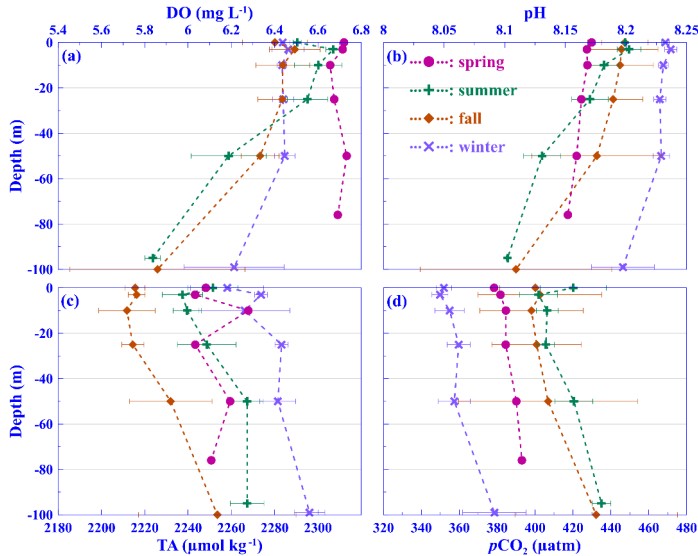


**Fig. 5.** Vertical profiles of mean (± SD) dissolved oxygen (DO; a), pH (b), total
alkalinity (TA; c), and $p$CO$_2$ (d) at station S10 in different seasons.

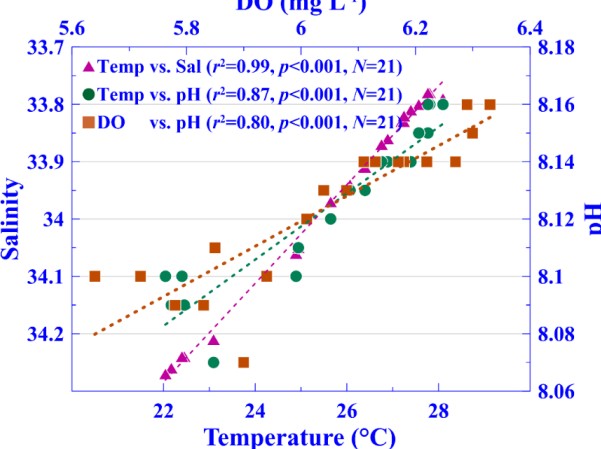


**Fig. 6** Relationships amongst temperature and salinity or pH, as well as dissolved
oxygen (DO) vs. pH, during multiple upwelling events at Station S1 (data from
Tew et al., 2014).





event, there is an intrusion of cold, low-DO, low-pH, and high-salinity deep-sea water
into the nearshore regions of Nanwan Bay (Fig. 6). Seawater property profiles of S10
provide additional evidence of upwelling, as indicated by the presence of low
temperatures (23.3±0.6℃), low pH (8.16±0.01), high salinity (34.32±0.03), and
relatively low $pCO_2$ (385.4±5.4 μatm) across the well-mixed water column in spring
(Figs. 3 & 5).

As temperature increases, $CO_2$ solubility decreases, causing an increase in $pCO_2$.

To accurately isolate and understand the specific impact of temperature variations on
$pCO_2$, it is crucial to normalize these factors. This approach allows for a clearer
distinction between temperature-induced changes and those driven by other
influences. Takahashi et al. (2002) proposed to evaluate the relative effects of
temperature and non-temperature effects on $pCO_2$ changes as follows:

$pCO_2 \text{ at } T_{obs} = (pCO_2)_{Mean\ annual} \times \exp[0.0423(T_{obs} - T_{mean})]$

$pCO_2 \text{ at } T_{mean} = (pCO_2)_{obs} \times \exp[0.0423(T_{mean} - T_{obs})]$

$pCO_2$ at $T_{obs}$ is calculated using the average $pCO_2$ to determine the $pCO_2$ value at a
given temperature; $pCO_2$ at $T_{mean}$ is the standardized $pCO_2$ value at the average
temperature; and $T_{mean}$ and $T_{obs}$ are the annual average temperature and the measured
temperature on-site, respectively. To assess the impact of temperature (T) and non-
temperature (nT) effects on $pCO_2$, the following equations were employed:



T effect = $pCO_2$ at $T_{obs}$ - $pCO_2$ at $T_{mean}$

nT effect = $pCO_2$ - $pCO_2$ at $T_{obs}$

The fluctuations in the mean $pCO_2$ at each monitoring station over time suggest that
temperature and non-temperature effects had distinct influences on the average $pCO_2$
at each station (Fig. 7). This means that seasonal changes in $pCO_2$ are influenced by
both temperature and non-temperature effects (e.g., gas exchange, tides, currents,
river discharge, upwelling, vertical mixing, and biological processes), with some
stations showing larger changes than others. It is believed that the stations with larger
$pCO_2$ variations are likely dominated by either temperature or non-temperature

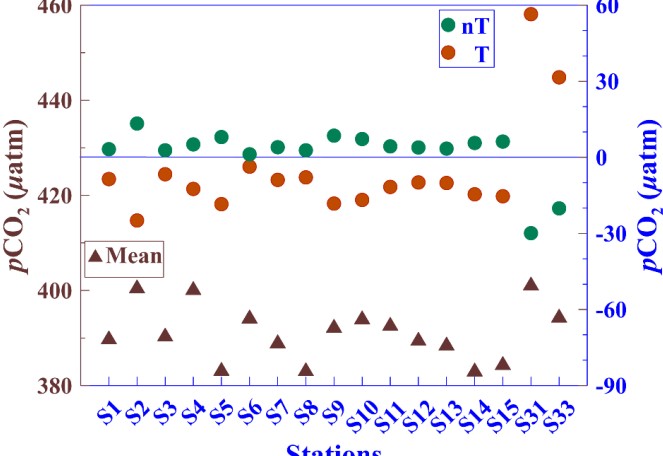


Fig. 7. Mean values and impact levels of surface water $pCO_2$ at each station in

Nanwan Bay are presented. The "Mean" represents the average value for each

station across the four seasons and is plotted on the left y-axis. The terms "nT"

and "T" refer to the non-temperature and temperature effects on surface water

$pCO_2$, respectively, and are displayed on the right y-axis for clarity.



344 effects, while the smaller changes reflect the mutual offsetting of the two effects (Dai

345 et al., 2009). For example, the variability in $pCO_2$ observed at S31 and S33, which are

346 located near the Nuclear Power Plant outlet, is likely driven by temperature change, as

347 the water temperature in this area was consistently higher and exhibited greater

348 variability compared to the surrounding area throughout the year. In fact, we expected

349 that temperature effects on $pCO_2$ would be more pronounced at these sites.

350  During the entire study duration, variation in surface water $pCO_2$ in Nanwan

351 Bay was influenced by a combination of temperature and non-temperature factors

352 throughout all seasons, albeit with varying degrees of influence in different seasons.

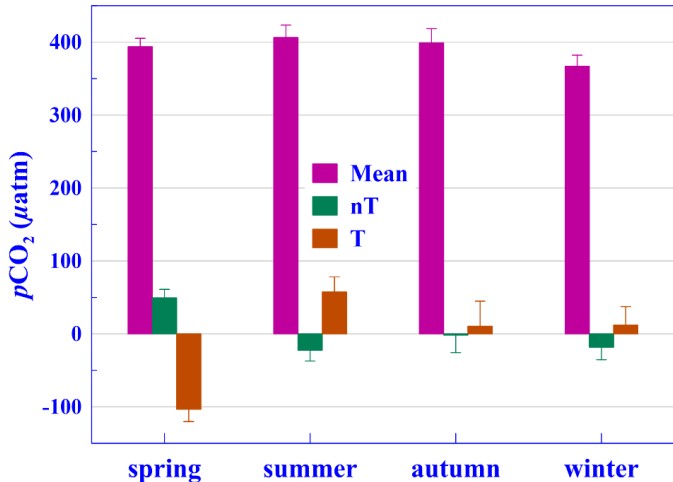

354 **Fig. 8.** Mean values and impact levels of surface water $pCO_2$ in Nanwan Bay during

355  different seasons. "Mean" represents the average value across sampling stations

356  for each season. "nT" denotes non-temperature effects on surface water $pCO_2$,

357  while "T" signifies temperature effects on surface water $pCO_2$. Error bars

358  represent standard deviation.



Notably, the effect of temperature was more prominent in the spring and summer (Fig.
8). The relationship between surface water $p\mathrm{CO_2}$, surface water temperature, and Chl
$a$ concentration revealed a significant correlation with $p\mathrm{CO_2}$ and temperature in the
summer ($p<0.01$; Fig. 9a), and a positive correlation between $p\mathrm{CO_2}$ and Chl $a$ in
autumn ($p<0.05$; Fig. 9b). This suggests that temperature and Chl $a$ may be the factors
affecting surface water $p\mathrm{CO_2}$ in summer and autumn, respectively. In general, Chl $a$
affects $p\mathrm{CO_2}$ by driving photosynthesis, which removes $\mathrm{CO_2}$ from seawater via

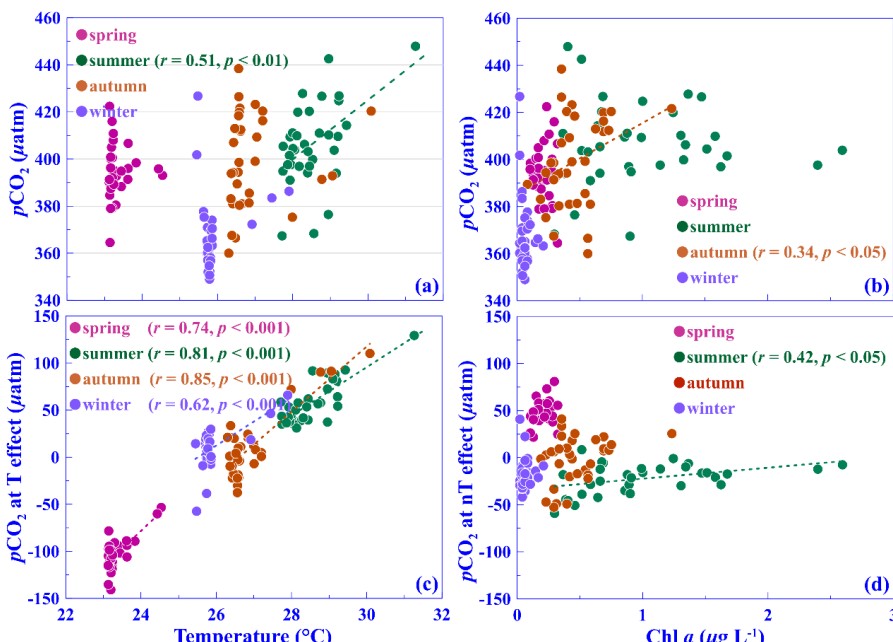


**Fig. 9.** Relationships between surface water $p\mathrm{CO_2}$ and (a) temperature, and (b) Chl $a$,

(c) the temperature (T) effect on surface water $p\mathrm{CO_2}$ and temperature, and (d)

the non-temperature (nT) effect on surface water $p\mathrm{CO_2}$ and Chl $a$ across

different seasons in Nanwan Bay. The linear relationships, along with the

corresponding r and p values, are provided for reference.



phytoplankton activity (Chen et al., 2019). Higher chlorophyll levels suggest
increased primary productivity, leading to greater $CO_2$ drawdowns during the day.

To further assess the influence of T and nT effects on surface water $p$CO$_2$ across

different seasons, we compared T-driven surface water $p$CO$_2$ with temperature and
nT-driven surface water $p$CO$_2$ with Chl $a$ (one of the nT factors; Fig. 9c, d).
Significant correlations were observed between T-driven surface water $p$CO$_2$ and
temperature in all seasons (Fig. 9c), and between nT-driven surface water $p$CO$_2$ and
Chl $a$ during summer (Fig. 9d). Overall, the results (Figs. 8 and 9) indicate that
temperature is the primary driver of seasonal $p$CO$_2$ variation, with non-temperature
factors, particularly Chl $a$, also contributing. However, the lower $r$ values suggest that
additional, unmeasured factors may be influencing the temporal variation in $p$CO$_2$
(Fig. 9d).

As mentioned above, seawater $p$CO$_2$ levels can be influenced by phytoplankton

via photosynthesis. Therefore, nutrient availability in seawater primarily affects $p$CO$_2$
levels by either promoting or limiting phytoplankton growth and consequently
primary production. In the case of Nanwan Bay, like many coral reef ecosystems, its
benthic environment supports nutrient regeneration through processes such as organic
carbon decomposition and other processes. However, due to the high shallow water
temperature and frequent stratification, regenerative nutrients cannot easily be

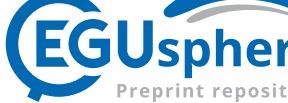

transported to the shallows. This results in the shallow areas rarely becoming
eutrophic (Leichter et al., 1996; Torréton, 1999; Wolanski and Pickard, 1983).
Another reason for Nanwan Bay's oligotrophy is that when nutrients flow into reef
areas, resident organisms may quickly utilize them (Wilkerson and Trench, 1986).
Although nutrient input from outside the bay is greater than the outward flux, rapid
circulation of water in Nanwan Bay leads to unused nutrients being swiftly exported
out of the bay (Su, 2009). This causes oligotrophy and high benthic productivity in the
area. Su (2009) reported that during spring tides, the water can be replaced in just 1.6
tidal cycles. Therefore, nutrient levels and Chl *a* may have only small influences on
$p$CO$_2$ in Nanwan Bay, with temperature changes and seawater movement having a
more significant impact.

**3.3 Spatial and temporal variations of △$p$CO$_2$ and CO$_2$ air-sea flux.** The

partial pressure difference between CO$_2$ in surface seawater and the atmosphere,
denoted as △$p$CO$_2$, indicates the direction of air-sea CO$_2$ exchange. When △$p$CO$_2$>0,
CO$_2$ in seawater is released into the atmosphere, contributing to an increase in
atmospheric CO$_2$ concentration (i.e., a source). On the other hand, when △$p$CO$_2$<0,
CO$_2$ from the atmosphere enters the seawater, acting as a sink for atmospheric CO$_2$. In
spring, the △$p$CO$_2$ range was between -14.3 and 27.7 µatm, with an average of 7.7
(±10.8) µatm (Fig. 10a). The highest value was observed near the Nuclear Power



Plant outlet station, i.e., St. 2. In summer, it ranged between -5.1 and 54.5 µatm, with
an average of 29.3 (±16.1) µatm. The highest value was measured near station S7. In
autumn, the range was between -14.6 and 51.2 µatm, with an average of 21.2 (±18.6)
µatm. The highest value was observed near stations S10-S12. In winter, the range was
between -33.2 and 31.3 µatm, with an average of -16.1 (±14.5) µatm. The highest
value occurred near stations S2. It is important to note that $p\text{CO}_2$ measurements in this
study were limited to daytime, when photosynthesis is actively occurring, typically
resulting in lower $p\text{CO}_2$ levels. As reported in this study and other coral reef
ecosystems (Yan et al., 2018) , significant diurnal variations in seawater $p\text{CO}_2$ have
been observed. Therefore, it is likely that the $\triangle p\text{CO}_2$ values presented here may be
underestimates due to this limitation.

Based on data from the Central Weather Bureau's Oluanpi buoy, the average

sampling date wind speed during the southwest monsoon season (summer) was 1.4
(±1.0) m s$^{-1}$, while during the northeast monsoon seasons (spring, autumn, & winter),
it was 10.6 (±0.8), 9.2 (±2.5), and 3.3 (±0.7) m s$^{-1}$, respectively. In other words, high
wind speeds are consistently observed during the northeast monsoon, in contrast to
the relatively lower wind speeds experienced during the summer season along the
coast of Taiwan (Ren et al., 2022). Utilizing these wind speed values, the $\text{CO}_2$ air-sea
exchange flux in Nanwan Bay was calculated (Figure 10b). During spring, the $\text{CO}_2$



flux ranged from -19.9 to 32.6 mmol m$^{-2}$ day$^{-1}$ (average=8.2±12.7 mmol m$^{-2}$ day$^{-1}$). In
summer, the $CO_2$ flux ranged from 0.0 to 4.2 mmol m$^{-2}$ day$^{-1}$ (average=0.9±1.2 mmol
m$^{-2}$ day$^{-1}$). During autumn, the $CO_2$ flux ranged from -17.1 to 59.5 mmol m$^{-2}$ day$^{-1}$,
with an average of 16.0±18.4 mmol m$^{-2}$ day$^{-1}$. Finally, in winter, the $CO_2$ flux ranged
from -5.9 to 5.6 mmol m$^{-2}$ day$^{-1}$, with an average of -1.8±2.4 mmol m$^{-2}$ day$^{-1}$. These
findings highlight that wind speed plays a crucial factor in regulating $CO_2$ air-sea
exchange flux. Moreover, any factors that impact wind speed can significantly affect
gas exchange estimates. For instance, when using daily wind speed data for the

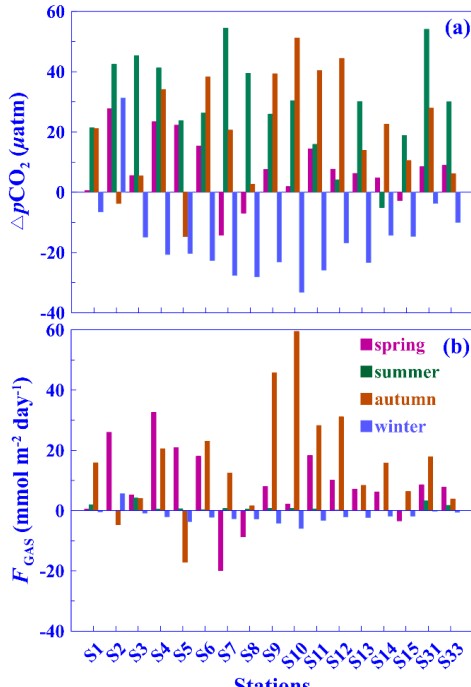


**Fig. 10.** Seasonal variation in (a) surface water $\triangle pCO_2$ and (b) air-sea $CO_2$ exchange

flux ($F_{GAS}$) at each station. Values are presented relative to the annual mean

(scaled to 0).





sampling month, the $CO_2$ flux increase from 0.9 ($\pm$1.2) to 12.4 ($\pm$9.9) mmol m$^{-2}$ day$^{-1}$
in summer, and 16.0 ($\pm$18.4) to 29.5 ($\pm$11.9) mmol m$^{-2}$ day$^{-1}$ in fall (Table S2).

The area between Cape Moubitou and Cape Oluanpi, covering approximately 30

km² (Fig. 1), shows $CO_2$ fluxes of 6.2 tC, 0.7 tC, 11.9 tC, and -1.3 tC in spring,
summer, fall, and winter, respectively, based on wind speeds recorded on the sampling
date (Table S2). However, when using daily wind speed data for the sampling month,
significant changes in $CO_2$ flux are observed, such as an increase from 0.7 to 9.4 tC in
summer and from 11.9 to 21.9 tC in fall (Table S2). This indicates that $CO_2$ flux
estimates are highly sensitive to wind speed variability, introducing uncertainty in
determining whether Nanwan Bay's coral reef ecosystem functions as a carbon sink or
source based on this limited dataset. Most coral reef areas act as atmospheric $CO_2$
sources, such as Bermuda (14.4 gC m$^{-2}$ year$^{-1}$; Bates et al., 2001), Okinawa (21.6 gC
m$^{-2}$ year$^{-1}$; Ohde and Van Woesik, 1999), the Great Barrier Reef (18 gC m$^{-2}$ year$^{-1}$;
Frankignoulle et al., 1996), French Polynesia (1.2 gC m$^{-2}$ year$^{-1}$; Frankignoulle et al.,
1996; Gattuso et al., 1997; Gattuso et al., 1993), and Hawaii (17.4 gC m$^{-2}$ year$^{-1}$;
Fagan and Mackenzie, 2007). In addition to wind speed, the primary factor
influencing the atmospheric carbon sink/source nature of this coral reef ecosystem is
the substantial vertical mixing and upwelling observed during the spring and winter.
These periods are characterized by a well-mixed water column, lower seawater





temperatures, and reduced $pCO_2$ levels, particularly during the northeast monsoon,
when high winds significantly enhance the $CO_2$ sea-air gas exchange flux. In contrast,
despite high  $\triangle pCO_2$ values in summer, wind speeds were relatively low, especially
during the southwest monsoon, making the $CO_2$ sea-air flux insufficient to offset the
carbon sink observed in winter. Although land-based inputs can affect nearshore
environments, leading to $pCO_2$ variations and influencing $CO_2$ sea-air flux (Meng et
al., 2008; De Carlo et al., 2007), the impact here is minimal, as no large river is near
this coral reef ecosystem. Additionally, the wind speed data used for $CO_2$ flux
calculations were collected from buoys within the ecosystem, providing an accurate
reflection of local conditions. It is important to note that short-term and long-term
wind speed fluctuations, prevailing climate conditions, and specific events can affect
$CO_2$ flux calculations, given the nonlinear relationship between wind speed and gas
exchange (Chou et al., 2011; Evans et al., 2012; De La Paz et al., 2011).

**4.  Conclusions**

Nanwan Bay experiences notable seasonal variations in temperature and salinity,

largely influenced by the South China Sea and the Kuroshio Current. These changes
impact the seawater carbonate system (including $pCO_2$), with additional influences
from vertical water movement and biological activity. Temperature emerged as the
primary driver of spatio-temporal variations in $pCO_2$, particularly at the consistently



warmer outlet station. Non-temperature factors also played a role during the spring,
while the interaction between temperature and other factors became more prominent
in the autumn and winter. During the winter, the bay absorbs more $CO_2$ from the
atmosphere, whereas in the spring, summer, and autumn, it releases more $CO_2$ than it
absorbs. The complex interplay of temperature, water mass origin, vertical water
movement, and biological activity in Nanwan Bay significantly affects its carbon
dioxide dynamics and its influence on atmospheric $CO_2$ levels.
**5. Acknowledgements**

This study was supported by grants from the Ministry of Science and Technology

of Taiwan (MOST) and National Science and Technology Council (NSTC) of Taiwan
through various grants, specifically: MOST 111-2611-M-259-002, MOST 110-2611-
M-259-002, MOST 109-2611-M-259-003, MOST 108-2611-M-291-005, MOST 107-
2611-M-291-001, and MOST 106-2611-M-291-006 awarded to PJM. CCC was also
supported by NSTC 112-2611-M-003-004. and NSTC 113-2611-M-003-002.
Data were submitted to Dryad for archiving (doi: 10.5061/dryad.63xsj3v7d).
**6. Credit author statement**

This manuscript was conceptualized by PJM and CCC; CMC and HYH conducted

investigations on all cruises and collected and analyzed the initial data; PJM, ABM,



and CCC wrote the initial draft; all authors provided comments and edits. The authors
declare that they have no competing interests.



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
