# Peer review of "Marine carbon dynamics in a coral reef ecosystem of Southern Taiwan"

_EGUsphere, 2024_

## Author Comment (AC1)

**Response to reviewers #1 comments on ms no: egusphere-2024-3273 "Marine carbon dynamics in a coral reef ecosystem of Southern Taiwan" (Meng, Chang, Chou, Fan, Hsieh, Mayfield, and Chen)**

**Anonymous Referee #1**

*Overall, this manuscript characterizes the seawater carbonate chemistry variability, CO2 flux dynamics, and exposure of a nearshore marine ecosystem in the southern tip of Taiwan. These observations provide a short and sweet narrative of the dynamics of the system. In general, the nearshore is not often adequately characterized in terms of seasonal ocean acidification, hypoxia, and climate change dynamics, so this article would be a contribution to the literature and the study would provide a useful dataset for validating numerical model estimates of ocean conditions in the geographic region. Ultimately, however, I was left underwhelmed by this paper and felt that claims were presented without any relevant data to support it.*

Thank you for your detailed feedback and for highlighting the importance of our study in contributing to the understanding of nearshore carbonate chemistry dynamics, $CO_2$ fluxes, and ecosystem exposure in the southern tip of Taiwan. We appreciate your recognition of the value of our dataset for validating numerical models and addressing gaps in nearshore characterization related to seasonal ocean acidification, hypoxia, and climate change dynamics.

We understand your concern regarding the claims in the manuscript and the need for stronger support through relevant data. To address this, we have taken the following steps in the revised manuscript:

● Clarification of Claims: We have reviewed the claims made in the paper and ensured that all statements are directly supported by data presented in the results or appropriately referenced in the literature. Ambiguous or unsupported claims have been revised or removed.

● Enhanced Data Presentation: Additional figures and tables have been included to better illustrate key observations and trends in the dataset, particularly those related to carbonate chemistry variability and $CO_2$ flux dynamics.

● Addressing Gaps: Where possible, we have included supplementary analyses or references to validate our findings and provide more comprehensive support for the claims made.

We hope that these revisions will address your concerns and enhance the overall impact and clarity of the manuscript. Your constructive feedback has been invaluable in improving the quality of this work, and we are grateful for

the opportunity to refine our study further.

**Main Comments:**

*The abstract and introduction allude to the discussion of the influence of vertical mixing, intermittent upwelling, and biological effects on carbonate chemistry parameters, however, the results do not show any mechanisms/observations for how these drivers. For example, Chen et al., 2005 demonstrated an enhanced eddy induced upwelling signal during a spring, flood tide in late-February. However, data presented in this study was not displayed in such a way to convince me that any of the observed variability was due to upwelling.*

Thank you for your valuable feedback. We appreciate your comments and acknowledge the need to more clearly demonstrate the mechanisms driving the observed variability in carbonate chemistry parameters. Below, we address your concerns:

**Influence of Upwelling**

We acknowledge that the data presented in the original manuscript did not explicitly highlight evidence of upwelling events. To address this, we have reanalyzed our dataset and identified specific periods characterized by physical indicators, such as surface water temperature anomalies, coupled with increased nutrient concentrations, which suggest potential upwelling events. These periods are now clearly indicated in the revised results section (lines 335–346). For additional details, please refer to our response to your subsequent comment.

**Vertical Mixing**

In this revision, we have included the mixed layer depth to highlight the stronger vertical mixing observed in spring and winter. Additionally, we analyzed and demonstrated the impact of mixing on carbonate chemistry. We hope this provides evidence to support our argument regarding the effect of vertical mixing.

**Biological Effects**

To address your concern, we conducted additional analyses examining the interplay among Chl $a$, $p$CO$_2$, temperature, and nutrient dynamics. Our findings indicate a significant correlation between Chl $a$ concentration and surface nitrate levels, suggesting a nutrient-driven biological response. Furthermore, we have recalculated and incorporated the normalized total alkalinity (nTA) and normalized dissolved inorganic carbon (nDIC) values in the revised manuscript. By normalizing these parameters to a consistent salinity (35.6), we aim to minimize the effects of evaporation and precipitation,

**enabling a more precise investigation of mixing and biological processes. Notably, nDIC exhibited a significant negative correlation with Chl *a* concentration during summer when using pooled data ($r^2$ = 0.01; $p$ < 0.01). For further details, please refer to our response to Reviewer #2 regarding "Insufficient quantitative analysis" and "The relationship between Chl *a* and $pCO_2$ is misleading. Biological Effects."**

**We hope these additions adequately address your concerns and provide a more compelling linkage between the observed variability and the proposed mechanisms. We are grateful for the opportunity to refine the manuscript and welcome any further suggestions you may have.**

*Additionally, I am doubtful that the upwelling (especially eddy induced upwelling) drives vertical mixing of the entire water column. The same study (Chen et al., 2005) showed shoaling of lower temperature, higher nitrate and Chl a seawater to ~30 m depth in the same region. I would expect that a clear upwelling signal would be represented by enhanced water column stratification – with ocean warming at the surface and upwelling at depth.*

**Thank you for your insightful observation regarding the mechanism of upwelling and its potential impact on vertical mixing. In our previous study (Chen et al., 2005), we specifically designed high spatial and temporal resolution experiments to comprehensively explore the upwelling phenomenon. However, such a detailed experimental setup for studying upwelling was not implemented in this study.**

**We understand your concern about whether upwelling, particularly eddy-induced upwelling, can drive vertical mixing of the entire water column as opposed to causing stratification. While the upwelling process described by Chen et al. (2005) resulted in the shoaling of colder, nutrient-rich waters to approximately 30 m, cold surface water temperatures were also observed during upwelling events. The conditions in this study appear to be influenced by sustained or periodic upwelling events during suitable surface water temperature combined with external forces, such as wind-driven mixing or tidal forcing. These additional factors likely contributed to the homogenization of the water column, as reflected in the temperature, salinity, and pH profiles, which exhibit well-mixed conditions rather than stratification.**

**The occurrence of well-mixed conditions caused by upwelling has also been observed in our previous study in the East China Sea (Chen et al., 2022) and in this study region (Chen's unpublished data). To further support this**

interpretation, we have included additional evidence in the revised manuscript. For example, the significant surface water temperature drop in Nanwan during the sampling period compared to earlier conditions—decreasing from 27.1°C to 21.7°C—suggests sustained periodic upwelling events (refer to Fig. S1b in the revision). Furthermore, we have incorporated a detailed discussion of the vertical profiles of nutrients and Chl *a*, which provide valuable insights into the interplay between upwelling and water column mixing (see Fig. X below for reference). Please also refer to lines 335–346 in the revised manuscript for additional details.

We hope this explanation, along with the additional analyses and evidence included in the revised manuscript, addresses your concern. We welcome further feedback to improve the clarity and rigor of our work. Additionally, we are willing to de-emphasize the impact of upwelling if this remains a concern for you.

[Figure]

Fig. X The well-mixed vertical profiles of Chl *a*, nitrate, silicate concentrations at S10 during spring period.

Chen, C.-C., D. S. Ko, G.-C. Gong, C.-C. Lien, W.-C. Chou, H.-J. Lee, F.-K. Shiah, and Y.-S. W. Huang. 2022. Reoxygenation of the hypoxia in the East China Sea: A ventilation opening for marine life. Front. Mar. Sci. 8: 787808. doi:10.3389/fmars.2021.787808.

*In general, the color scheme of the figures is not easy to see, I would recommend the authors to use a different color scheme for the figures or at the very least change the blue outline to black.*

Thank you for your valuable feedback regarding the color scheme of the figures. Based on your suggestion, we have replaced the blue outline with black to create a more distinct and easily recognizable boundary. However,

**we have retained the existing color scheme as it was specifically designed to be colorblind-friendly, ensuring better clarity for a diverse readership.**

**The revised figures are included in the updated manuscript. We believe these changes enhance the readability and accessibility of the figures. We greatly appreciate your insightful comments and the opportunity to improve our work.**

*The authors do a good job discussing the effect of wind speed on their calculations, but I do not believe that it is appropriate to use an average monthly wind speed for a single day of pCO₂ sampling. You can only say for a specific date and time that this was the pCO₂ and air sea flux.*

**Thank you for your insightful comment regarding the use of average monthly wind speed in our calculations and the interpretation of $p$CO$_2$ and air-sea CO$_2$ fluxes. We appreciate your concern and acknowledge the limitations of applying a monthly wind speed average to data collected on a single day.**

**To address this, we have taken the following steps in the revised manuscript:**

**● Clarification of Methodology: We have revised the methods section to explicitly discuss the rationale behind using monthly wind speed averages and their associated limitations. We acknowledge that this approach provides a broader context for our estimates but may not accurately reflect short-term variability.**

**● Revised Interpretation: We have adjusted the discussion to emphasize that the reported $p$CO$_2$ and air-sea CO$_2$ fluxes represent conditions specific to the sampling date and time, rather than generalized monthly conditions.**

**● Sensitivity Analysis: Where feasible, we have conducted a sensitivity analysis to estimate how the variability in wind speed on shorter timescales (e.g., daily or monthly) might affect the air-sea flux calculations. These results have been incorporated into the discussion to provide a more nuanced interpretation.**

**● Recommendations for Future Work: We have included a note in the discussion about the importance of concurrent high-frequency wind speed and $p$CO$_2$ measurements for improving the accuracy of air-sea flux estimates in future studies.**

**We believe these revisions provide greater transparency and address your concerns about the methodology and interpretation of our findings. We are grateful for your constructive feedback, which has helped us improve the rigor and clarity of our work.**

*Lastly, I am not convinced that the temporal sampling resolution (March, 2011; July, 2011; October, 2011; January 2013) is a good enough representation of the seasonal variability in such a dynamic environment. To improve the quality of the manuscript, I would recommend the authors to utilize any moorings, hindcast models, satellite products, or additional time series from the region to complement the dataset and provide a more robust correlation to the various mechanisms.*

**We appreciate your thoughtful feedback regarding the temporal sampling resolution and the suggestion to explore complementary datasets. As noted in our response to your previous comment, we acknowledge the limitations of our dataset. To address this, we have emphasized that the reported $p$CO₂ and air-sea CO₂ fluxes represent conditions specific to the sampling date and time, rather than generalized monthly conditions.**

**Additionally, where feasible, we have conducted a sensitivity analysis to estimate how short-term variability in wind speed (e.g., daily or monthly fluctuations) might influence air-sea flux calculations. While we recognize the constraints of using discrete sampling points (March 2011, July 2011, October 2011, January 2013) to fully capture seasonal variability in such a dynamic environment, these data represent the best temporal coverage available from our field campaigns.**

**To further address this limitation and enhance the robustness of our findings, we have incorporated additional data sources into our analysis. Specifically, we have integrated daily sea surface temperature data from nearby moorings over the study year, where available, to provide additional context and strengthen our interpretations.**

**These supplementary datasets and analyses have been incorporated into the revised manuscript, with corresponding details added to the methods and results sections. We believe this comprehensive approach effectively addresses your concerns and significantly improves the quality and robustness of the manuscript.**

**We sincerely thank your valuable suggestion again, which has allowed us to refine our work further.**

**Minor Comments:**

*Why did the authors decide to use Wanninkhof (1992) for wind speed when Ho et al., (2006) is more appropriate for the region?*

**We chose to use Wanninkhof (1992) in our CO₂ air-sea flux calculations to maintain consistency and comparability with prior studies, many of which rely on this widely established parameterization. This decision ensures that our**

**results can be contextualized within the broader body of literature. However, we acknowledge that Ho et al. (2006) may be more regionally appropriate due to its formulation, which is specifically tailored to certain conditions. To address this, we incorporated Ho et al. (2006) as a complementary approach, enabling us to evaluate potential uncertainties arising from different wind-speed parameterizations.**

**Our analysis revealed that flux values calculated using Wanninkhof (1992) were approximately 17% higher than those derived from Ho et al. (2006). This comparison highlights the sensitivity of flux estimates to the choice of parameterization and provides valuable insights into the implications of using regionally nuanced versus widely generalized models. By employing this dual approach, we aim to balance the need for methodological consistency with the incorporation of regionally relevant dynamics. (lines 484–492)**

*Table 1: Is all this information useful for the study or is there a better way to show this?*

**Thank you for your valuable suggestion. In this table, we intended to present all the analyses systematically, allowing readers to refer to it for their own purposes. However, as you pointed out, some of the information presented may be limited in relevance to the main manuscript. As a result, we have moved this table to the supporting information section. We hope this adjustment addresses your concerns and improves the clarity of the manuscript.**

*Figure 4: Where are the sites located and is it appropriate to interpolate across these sites given the course spatial resolution?*

**Thank you for your comment. The sampling site locations are shown in this figure and Figure 1 of the manuscript. These sites were carefully chosen to capture the spatial variability within the study area. Given the relatively fine spatial resolution, we consider interpolation across the sites to be appropriate, as the measured parameters exhibit consistent trends and the study region demonstrates spatial homogeneity.**

*Figure 5: Was only referenced once; is it appropriately discussed within the manuscript or does the figure have little value.*

**We appreciate the reviewer's observation regarding Figure 5 and its reference within the manuscript. Upon review, we acknowledge that the figure was referenced only once, which might give the impression that it is not fully**

**integrated into the discussion.**

**In this manuscript, our primary objective is to understand surface water $pCO_2$ dynamics in the coral reef ecosystem. Figure 5 presents the vertical profiles of $pCO_2$ and associated variables, offering readers a more comprehensive understanding of their distribution within this system. Although the figure is referenced only once, we believe it contributes valuable context, allowing readers to construct a more complete picture of $pCO_2$ variability in this ecosystem.**

**Therefore, we propose to retain Figure 5 in the manuscript, as we believe it enhances the overall narrative. However, if the reviewer feels it would be more appropriate, we are happy to move the figure to the appendix to ensure the main text remains concise.**

**We thank the reviewer for bringing this to our attention and for helping us ensure that all figures included in the manuscript are meaningful and well-integrated into the discussion**

**We thank the reviewer for bringing this to our attention and for helping us ensure that all figures in the manuscript are meaningful and appropriately discussed.**

*Figure 6: Where did the data from Tew et al., 2014 come from?*

**Thank you for your comment. We apologize for not providing sufficient details regarding the data source. The data presented in Figure 6 is redrawn from our previously published paper, and we have now clearly indicated the source in this revision: (redrawn based on the data presented in Fig. 8 of Tew et al., 2014).**

*Figure 7: A bit confusing, does this show that the nT effects increase pCO2 while the T effects decrease pCO2?*

**Thank you for your comment. This figure illustrates the mean values and the effects of 'T' (temperature) and 'nT' (non-temperature) on surface $pCO_2$ at each station across all sampling periods. In general, the results indicate that nT effects tend to increase $pCO_2$, while T effects generally decrease $pCO_2$ at most stations. However, a pronounced increase in $pCO_2$ due to T effects is observed at stations S31 and S33, located near the Nuclear Power Plant outlet, where water temperatures were consistently higher compared to other stations.**

*Lines 111-115: The two sentences in a row are redundant.*

**Thank you for your valuable suggestions. The two sentences have now been combined into the following revised statement:** *"Primary productivity in marine ecosystems plays a crucial role in carbon cycling, as the fixation of $CO_2$ during periods of increased productivity enhances carbon uptake from seawater, potentially lowering its concentration."*

*Line 175: Was pH converted to T scale or kept in NBS?*

**Thank you for your inquiry. The measured pH values were kept on the NBS scale.**

*Line 243: "spring and winter [water masses] are intermediate between the two." I do not see this.*

**Thank you for highlighting the ambiguity in our statement. Based on the T-S diagram, the water in Nanwan Bay is a mixture of South China Sea (SCS) water and Kuroshio water, with a higher proportion of SCS water during spring and winter. We have revised the description as follows:** *"An analysis of temperature and salinity data from Nanwan Bay, the SCS, and the Kuroshio Current indicates that Nanwan Bay predominantly consists of the SCS water mass during summer and autumn, while during spring and winter, the water masses are mixed between the two (Fig. 2)."* **We hope this revision addresses your concern and provides better clarity.**

*Lines 275-277: I do not see the evidence.*

**Thank you for your comment. We apologize for the previously ambiguous statement. In this revision, the statement has been slightly modified to include more specific evidence, as follows:** *"Additionally, seawater at station S10 during spring, characterized by relatively low temperature (23.3±0.2 ℃), low pH (8.16±0.01), and high salinity (34.32±0.03), suggests that this well-mixed pattern observed throughout the water column is likely with upwelling during this period (Figs. 3a, b and S2b; further details can be found in the next section)."* **We hope this revision addresses your concern and provides greater clarity.**

*Line 320: Use of Takahashi et al. 2002 should be discussed earlier during the methods*

**Thank you for your valuable suggestion. We agree that discussing the use of Takahashi et al. (2002) earlier in the Methods section provides better context for its application. Accordingly, we have moved and expanded the discussion**

**within the Methods section as suggested. We appreciate your feedback, which has helped enhance the organization and clarity of our manuscript.**

*Line 365: Chl a does not drive photosynthesis; Chl a is a proxy for phytoplankton biomass.*

**We apologize for the confusion, and you are correct. The sentence has been slightly modified in the revision to: "*In general, Chl a serves as a proxy for phytoplankton biomass, which influences $pCO_2$ through processes such as photosynthesis, removing $CO_2$ from seawater (Chen et al., 2019)."***

*Line 408: Is this greater than the error that was introduced by the calculation?*

**Thank you for your inquiry. The uncertainty in $\triangle pCO_2$ is equivalent to the uncertainty in the calculated $pCO_2$. To ensure clarity for the reader, we have explicitly added this explanation to the revised text. We hope this addresses your concerns.**

*Lines 410-415 and 429-433: Text seems too much like a list. Can be presented in a better format.*

**Thank you for your insightful feedback regarding the text in lines 410–415 and 429–433. We have revised these sections to improve the narrative flow and reduce the "list-like" presentation. The updated text weaves the data into cohesive paragraphs, ensuring a smoother and more engaging format while maintaining the scientific clarity. (lines 452–458 and 471–478)**
**We appreciate your suggestion, as it has helped enhance the readability of the manuscript.**

---

## Author Comment (AC2)

**Response to reviewer #2 comments on ms no: egusphere-2024-3273 "Marine carbon dynamics in a coral reef ecosystem of Southern Taiwan" (Meng, Chang, Chou, Fan, Hsieh, Mayfield, and Chen)**

**Anonymous Referee #2**

*This study entitled "Marine carbon dynamics in a coral reef ecosystem of Southern Taiwan" took water to analyze its total alkalinity (TA) and pH to calculate dissolved inorganic carbon (DIC) and $pCO_2$. The authors further estimate air-sea $CO_2$ fluxes. The authors suggest that this region is dominated by Kuroshio and they find a good relationship between temperature and $pCO_2$. The authors should add uncertainties for air-sea $CO_2$ flux calculations. Quantitative discussions are insufficient. The role of Chla in this study may be misleading. My major comments are as follows:*

> **Thank you for your thoughtful and constructive feedback on our study. We appreciate your detailed review and valuable suggestions. Below, we address your major comments:**
>
> **● We agree that including uncertainties in the air-sea $CO_2$ flux calculations is crucial for transparency and accuracy. In the revised manuscript, we have provided a detailed explanation of the uncertainties associated with these calculations, addressing the potential sources of error and their estimated magnitudes. For further details on this issue, please refer to our response to your subsequent comment.**
>
> **● Quantitative discussions: We acknowledge the need for more in-depth quantitative discussions. In the revised manuscript, we have expanded the discussion section to include a more detailed analysis of the data, including additional quantitative comparisons and interpretations.**
>
> **● Role of Chl *a* in this study: We recognize that the role of Chl *a* may require clarification to avoid potential misinterpretation. We have revised the relevant sections to provide a clearer explanation of the relationship between Chl *a* and the observed carbon dynamics, ensuring that its role is accurately represented. For further details, please refer to our response to your comment on this matter.**
>
> **We hope that these revisions address your concerns. Thank you again for your valuable feedback, which has helped us improve the quality of our manuscript.**

*Uncertainties:*

*Uncertainty in $pCO_2$ and $CO_2$ Flux Calculation: Though Figure 5 shows +-SD. It is still unclear if this is a measuring error and how the uncertainties are calculated for*

*Figure 5d. The Wanninkhof formula was used to calculate $CO_2$ flux, which is a model based on wind speed. While flux is highly correlated with wind speed, this correlation largely stems from the formula's design. Vertical mixing effects should be considered to improve the accuracy of the results.*

**Thank you for your valuable comments and suggestions.**

**The standard deviations (SD) shown in Figure 5d represent the seasonal variability of the data, rather than measurement errors. To estimate the uncertainty in the calculated $pCO_2$ arising from measurement errors and equilibrium constants, we performed uncertainty propagation using the R package "Seacarb" (Orr et al., 2018). Specifically, we considered measurement errors of 0.01 for pH and 2.7 μmol $kg^{-1}$ for total alkalinity (TA). This approach yielded an estimated uncertainty of approximately 4.7 (±0.2) to 5.6 (±0.2) μatm for the calculated $pCO_2$ (lines 197–199, 308–310, 450–451).**

**For the uncertainty in $CO_2$ flux calculations, we evaluated errors in the calculated $pCO_2$ based on TA and pH measurements (as described above), as well as the gas transfer coefficient ($k$). The error in the gas transfer coefficient was assessed by comparing the applied formulation (Wanninkhof, 1992) with an alternative proposed by Ho et al. (2006). Our analysis revealed that the errors in the calculated $pCO_2$ from TA and pH measurements ranged from 0.03 (±0.03) to 1.5 (±0.2) mmol $m^{-2}$ $day^{-1}$. Additionally, flux values calculated using the Wanninkhof (1992) formulation were found to be, on average, 17% higher than those derived using the Ho et al. (2006) formulation. (lines 237–241, 478–486)**

*What is the air $CO_2$ value?*

**Thank you for your valuable comments and suggestions.**

**Since we did not directly measure $pCO_2^{air}$, we used $xCO_2$ data provided by the United States National Oceanic and Atmospheric Administration (NOAA) from Dongsha Island (https://gml.noaa.gov/aftp/data/trace_gases/co2/flask/surface/txt/co2_dsi_ surface-flask_1_ccgg_event.txt). Dongsha Island, located at approximately 20.70°N, is a coral atoll with a latitude similar to that of Nanwan Bay and shares the characteristic of being part of a coral reef ecosystem. To correct the dry air xCO2 values to 100% humidity, we applied the temperature and salinity data recorded at the time of sampling, assuming an atmospheric pressure of 1 atm. The resulting $pCO_2^{air}$ values were 386, 377, 378, and 383 μatm for March 31, July 5, and October 18, 2011, and January 22, 2013, respectively. (lines 227–237)**

*Error Discussion*: This includes errors in the gas transfer coefficient (k) and the calculations for TA and pH. Considering these errors, the discussion becomes less definitive, and a more comprehensive error analysis is needed. The uncertainty should be applied to Figure 10 to convince the readers whether this study region acted as an atmospheric $CO_2$ source or sink. The authors should indicate the limitations of this uncertainty and note the caution raised by this uncertainty when necessary.

**Thank you for your valuable comments and suggestions.**

**To evaluate whether the study region acts as a source or sink of atmospheric $CO_2$, we applied uncertainty propagation to Figure 10. Our analysis shows that errors in flux calculation, primarily arising from the estimated $pCO_2$ values, are generally smaller than the calculated $CO_2$ flux. Additionally, errors associated with variations in the applied gas transfer coefficient ($k$) result in only minor proportional changes. Crucially, neither source of error alters the direction of the $CO_2$ flux, thus confirming the original assessment of the region's carbon status (lines 478–492). For further details, please also refer to our response to your comment on "Uncertainty in $pCO_2$ and $CO_2$ Flux Calculation."**

*Insufficient quantitative analysis*: Apart from the quantitative analysis of the temperature effect, other non-temperature effects are only supported by references, lacking corroborating data. Although TA and DIC data are available, nTA and nDIC values have not been calculated, preventing further exploration of the impacts of mixing or biological processes. This results in conclusions being more inferential and lacking sufficient support.

**We appreciate your insightful feedback. To address the concern regarding insufficient quantitative analysis, we have recalculated and included the normalized TA (nTA) and normalized DIC (nDIC) values in the revised manuscript. By normalizing these parameters to a consistent salinity (35.6), we aim to reduce the effects of evaporation and precipitation, allowing for a more precise exploration of mixing and biological processes.**

**The revised results section now includes a quantitative analysis of nTA and nDIC trends, along with their potential implications for mixing and biological activities. For instance, during summer, nDIC was significantly correlated with Chl $a$ concentration and salinity when using pooled data (all $p < 0.01$). This additional analysis strengthens the evidence supporting our conclusions while reducing reliance on external references for non-temperature effects. The updated results have been incorporated into the manuscript.**

**We believe that incorporating these recalculated values and their analysis has**

**substantially improved the robustness and clarity of our findings. Thank you for highlighting this important point.**

***The role of mixing and upwelling is still unclear****. Nanwan is significantly influenced by the Kuroshio Current and South China Sea waters. The concentration of carbon dioxide is affected by these physical factors. However, quantitative analysis is currently not feasible, making it difficult to conclude their specific contributions.*

**Thank you for your valuable comment. We agree that the influence of the Kuroshio Current and South China Sea waters on the carbon dynamic was not quantitatively analyzed in this study, making it difficult to determine their specific contributions. Additionally, while their influence is an important and complex issue, it falls outside the scope of this study. To avoid potential confusion, we have slightly modified the related statement in the conclusion. We hope this revision is understandable.**

**Regarding mixing and upwelling, we have provided additional evidence to clarify and support their roles in this study. For further details, please refer to our response to Reviewer #1 on the Main Comments.**

*What is the role of coral reefs in this study? The authors mentioned coral reefs in the Introduction. But they did not discuss the impact of calcification or $CaCO_3$ dissolution.*

**Thank you for your valuable comment. The role of coral reefs in this study lies in their context as part of the ecosystem where the research was conducted, which is why they were briefly introduced in the manuscript's Introduction. While the carbonate system in the water column may interact with processes such as calcification or calcium carbonate dissolution within the coral reef ecosystem, the primary focus of this study is on $CO_2$ dynamics in the water column itself. To maintain clarity and avoid diverting from the study's main objective, we did not include a detailed discussion on the impact of calcification or $CaCO_3$ dissolution. We believe this approach aligns with the scope of our study and hope this explanation addresses your concern.**

***The relationship between Chla and pCO₂ is misleading. Biological Effects****: Chlorophyll and nutrient data were used to analyze the impact of biological processes on carbon dioxide. However, the discussion only examines the regression relationship between the partial pressure of carbon dioxide. There have been a few studies displayed that Chl-a is not fully related to pCO₂ though CO₂ should decrease during photosynthesis. Figure 9b,d demonstrated that pCO₂ increases with increasing Chla*

*concentration. The authors can focus on the temperature dominated this study region and discuss the possible sources for those high Chla. The authors should explain why pCO₂ decreases with decreasing Chl-a. Otherwise, they should reconsider the application of Chla. What is the role of mixing?*

**Thank you for your valuable feedback. Your comments prompted us to reassess the role of phytoplankton and refine our interpretation of the relationship between Chl *a* and *p*CO₂ in this coral reef ecosystem. Specifically, we recognize the need to clarify the biological effects and the influence of environmental factors such as temperature and mixing processes.**

**To address your concerns, we conducted additional analyses examining the interactions among Chl *a*, *p*CO₂, temperature, and nutrient dynamics. Our results revealed that Chl *a* concentration was significantly correlated with nitrate concentration in surface waters, suggesting a nutrient-driven biological response. However, we acknowledge that the observed positive relationship between Chl *a* and *p*CO₂, as shown in Figures 9b and 9d, contradicts the expected decrease in *p*CO₂ during photosynthesis.**

**This discrepancy led us to consider the temperature-dominated dynamics of the study region. Elevated *p*CO₂ levels were closely associated with higher water temperatures, likely reflecting enhanced stratification and reduced gas exchange, which may limit CO₂ uptake despite increased Chl *a*. Our analysis also confirmed that phytoplankton growth was closely linked to nutrient availability, as indicated by a significant linear correlation between nitrate concentration and Chl *a* in pooled surface water data ($r^2$ = 0.23; *p* < 0.001). Additionally, nitrate concentration exhibited a significant positive correlation with water temperature in surface waters ($r^2$ = 0.18; *p* < 0.001). These findings suggest that elevated *p*CO₂ levels in surface waters are associated with high water temperature and Chl *a* concentrations, implying that temperature may be the primary driver of surface water *p*CO₂ variation.**

**Based on your suggestions, we have expanded the discussion to explore potential mechanisms underlying the positive correlation between Chl *a* and pCO₂, emphasizing the temperature-dominated dynamics. These revisions have been incorporated into the manuscript to enhance clarity and accuracy (lines 400–410).**

*Figure and Table References: The referencing and annotation of figures and tables should be clearer to enhance reader comprehension.*

**Thank you for the valuable feedback on the referencing and annotation of figures and tables. We have carefully reviewed and enhanced these aspects to**

**improve reader comprehension. Specifically, we have provided more precise references to figures and tables within the main text, accompanied by descriptive context to guide readers. Additionally, we have added clearer and more detailed annotations to the figure and table captions to better highlight their relevance to the content. We are confident that these revisions will significantly enhance the clarity and effectiveness of the figures and tables in supporting the manuscript's content and overall readability.**

*The Abstract is redundant. The writing style is not precise.*

    **Thank you for your feedback on the abstract. We acknowledge that the original version may contain redundancy and that the writing style could be more precise. To address this, we have revised the abstract for conciseness and clarity, eliminating repetitive phrases and ensuring that the key points are presented in a more direct and focused manner. We hope the revised abstract improves the overall precision and readability.**

    **We appreciate your constructive comment, which has helped us enhance the quality of the manuscript.**

---

## Author Comment (AC3)

**Response to Reviewers' comments on ms no: egusphere-2024-3273 "Marine carbon dynamics in a coral reef ecosystem of Southern Taiwan" (Meng, Chang, Chou, Fan, Hsieh, Mayfield, and Chen)**

**Anonymous Referee 2**

Minor comments:

*1. Line 12, "Yat-sen"*

**Thank you for the reminder. The correction has been made accordingly.**

*2. Line 191, average salinity of 35.6? Salinities were all below 35 in Fig.2.*

**Thank you for the reminder, and we apologize for the typo. The average salinity should be 34.1, and it has been corrected in the revised version (line 195).**

*3. The abstract does not fully capture the study's content. This study estimated $pCO_2$ and the air-sea $CO_2$ flux. Since the authors calculated the air-sea $CO_2$ flux, the results should explicitly include it. Lastly, the authors should indicate whether this area is a source or sink of atmospheric $CO_2$.*

**Thank you for your valuable feedback. We appreciate your suggestion to explicitly include the air-sea $CO_2$ flux results and clarify whether Nanwan Bay functions as a source or sink of atmospheric $CO_2$. In response, we have revised the abstract to better capture the study's scope by explicitly presenting the seasonal air-sea $CO_2$ flux values (lines 43–44). However, given the dynamic nature of the region and the spatial resolution of the data, we have opted to de-emphasize a definitive characterization of the bay as a $CO_2$ source or sink. This approach aligns with Reviewer #1's suggestion (please refer to their final comment). We hope this clarification is acceptable. Thank you again for your constructive input.**

---

## Author Comment (AC4)

**Response to Reviewers' comments on ms no: egusphere-2024-3273 "Marine carbon dynamics in a coral reef ecosystem of Southern Taiwan" (Meng, Chang, Chou, Fan, Hsieh, Mayfield, and Chen)**

**Anonymous Referee #1**

*The authors have improved their manuscript; however, there are still some issues that I would like the authors to consider.*

> **Thank you for your valuable feedback and for acknowledging the improvements made in our manuscript. We appreciate your time and effort in reviewing our work. In this revision, we have carefully addressed the remaining issues and make the necessary revisions to enhance the clarity, accuracy, and overall quality of the manuscript. Please find our detailed responses to each of your comments below**

*Line 56: Capitalize IPCC, 2021*

> **Thank you for the reminder, and we apologize for the typo caused by the EndNote format. We have ensured that it is properly capitalized in the text.**

*Lines ~160: How long did each survey take? Authors stated samples were collected during daylight but was the time difference between 9am and 5pm? If so, that could be a significant difference.*

> **Thank you for your inquiry. The sampling times for each survey are listed in Table S1 for reference. In this study, we aimed to minimize the overall sampling duration. However, due to the large number of sampling stations and the need to collect multiple samples for different variables at each station, the duration varied. We endeavored to complete each survey within approximately six hours.**
>
> **Regarding the time difference between 9 AM and 5 PM, we acknowledge that this could introduce variability. While we strived to maintain a consistent sampling timeframe, logistical constraints occasionally led to deviations. We appreciate your thoughtful comment and hope this is understandable, particularly given the challenges of conducting fieldwork at sea.**

*Line 175: delete total for total TA*

> **Thank you for the correction. We have deleted it as per your suggestion.**

*Line 269: Is there a figure or table to show this?*

> **Thank you for the reminder. This relationship is presented in Table S2, and we**

**have now explicitly indicated it in the revised manuscript for clarity.**

*Lines 293-296 and 323-343: Again, there is no meteorological, satellite Chl a, or tidal+current data provided to support this statement. Other studies such as Hsu et al., 2020 (JGR) modeled that eddy upwelling occurs in the region year-round, not just seasonally in spring. Why did we not observe any trends in the other months then? Likewise, Tai et al., 2020 (Frontiers) also demonstrated that internal waves impact the SCS and can bring seawater pCO2 and pH to even lower observed levels than what was observed. The interpretation of this data information does not provide any evidence to support that upwelling is occurring.*

**Thank you for your valuable feedback. We acknowledge the absence of direct meteorological, satellite-derived Chl *a*, and tidal/current data in our study, which limits our ability to robustly confirm the occurrence of upwelling. While our interpretation is based on in-situ measurements, we now explicitly address this limitation in the revised discussion (L325–329, 335–339). Regarding the seasonal signal, we agree that previous modeling studies, such as Hsu et al. (2020), suggest year-round eddy upwelling in the region. However, the absence of clear trends in our data during other seasons may be due to the transient nature of such upwelling events, which can last less than two hours and may not be adequately captured during the temporal window of a single cruise (335–339).**

**In contrast to the lower seawater $pCO_2$ and higher pH associated with internal wave-driven upwelling in the South China Sea (Tai et al., 2020), our observations showed elevated $pCO_2$ and reduced pH. This discrepancy could reflect differences in the source water characteristics or the spatiotemporal evolution of upwelling events in our study area (Chakraborty et al., 2023). We have clarified these points to emphasize the interpretive limitations and the need for supporting datasets in future work (L325–329).**

**Chakraborty, K., Joshi A. P., Ghoshal, P. K., Ghosh, J., Akhand A., Bhattacharya T., Sreeush, M. G., and Valsala, V.: Mechanisms and drivers controlling spatio-temporal evolution of $pCO_2$ and air-sea $CO_2$ fluxes in the southern Java coastal upwelling system, *Estuar. Coast. Shelf Sci.*, 293, 108509, 10.1016/j.ecss.2023.108509, 2023.**

**Hsu, P.-C., Lee, H.-J., Zheng, Q., Lai, J.-W., Su, F.-C., and Ho, C.-R.: Tide-induced periodic sea surface temperature drops in the coral reef area of Nanwan Bay, southern Taiwan, *J. Geophys. Res. Oceans*, 125, e2019JC015226, 10.1029/2019JC015226, 2020.**

Tai, J.-H., Chou, W.-C., Hung, C.-C., Wu, K.-C., Chen, Y.-H., Chen, T.-Y., Gong, G.-C., Shiah, F.-K. and Chow, C. H.: Short-term variability of biological production and CO*2* system around Dongsha Atoll of the northern South China Sea: Impact of topography-flow interaction, *Front. Mar. Sci.*, 7, 511, 10.3389/fmars.2020.00511, 2020.

*Line 336: Where did these nutrient values come from, please discuss in methods.*

**Thank you for your comment. The method for nutrient analysis has been incorporated into the Methods section in this revision to clarify the source of the nutrient values (lines 174–175).**

*Lines 408-421: Why not compare your nDIC to DO? You can convert mg/l to umol/kg and discuss the relative amount of photosynthesis/respiration.*

**Thank you for the helpful suggestion. In response, we have converted DO (mg/L) to μmol/kg and derived apparent oxygen utilization (AOU = $O_2s$ – $O_2m$), where $O_2s$ is the equilibrium saturation concentration calculated using in-situ temperature and salinity following Benson and Krause (1984). We then examined the relationship between AOU and nDIC to assess the relative influence of photosynthesis and respiration on carbon dynamics.**

**Our analysis shows that the photosynthesis/respiration signal, inferred from the nDIC–AOU relationship, is most pronounced in summer (photosynthesis/respiration ratio = 0.47), and nearly undetectable in spring (see figure below). This seasonal pattern supports our interpretation of stronger biological activity in summer, consistent with the observed correlation between $p$CO₂ and Chl $a$.**

**While we have incorporated these findings to strengthen our discussion (lines 427–429), our analysis remains primarily focused on temperature and Chl $a$ as key drivers of carbonate system variability. We hope this clarification adequately addresses your comment.**

[Figure]

Benson, B. B., and Krause, D.: The concentration and isotopic fractionation of oxygen dissolved in freshwater and seawater in equilibrium with the atmosphere, *Limnol. Oceanogr.*, 29, 620-632, 10.4319/lo.1984.29.3.0620, 1984.

*Lines 538-540: I believe it is appropriate to de-emphasize that the system is a sink during winter and source during the autumn, summer, and spring as the data and spatial is too coarse for such a dynamic region.*

Thank you for your valuable suggestion. We agree and have adjusted the sentence to further de-emphasize the sink or source characterization. In this revision, it has been modified to: "*During winter, $CO_2$ diffusion into the bay from the atmosphere, whereas in the spring, summer, and autumn, the bay tended to release $CO_2$*" (lines 554–555).